# Current Concepts in the Management of Primary Lymphedema

**DOI:** 10.3390/medicina59050894

**Published:** 2023-05-06

**Authors:** Jenna-Lynn B. Senger, Rohini L. Kadle, Roman J. Skoracki

**Affiliations:** Wexner Medical Center, Department of Plastic Surgery, Ohio State University, Columbus, OH 43210, USA

**Keywords:** primary lymphedema, congenital lymphedema, lymphovenous bypass, lymphovenous anastomosis, vascularized lymph node transfer

## Abstract

Primary lymphedema is a heterogeneous group of conditions encompassing all lymphatic anomalies that result in lymphatic swelling. Primary lymphedema can be difficult to diagnose, and diagnosis is often delayed. As opposed to secondary lymphedema, primary lymphedema has an unpredictable disease course, often progressing more slowly. Primary lymphedema can be associated with various genetic syndromes or can be idiopathic. Diagnosis is often clinical, although imaging can be a helpful adjunct. The literature on treating primary lymphedema is limited, and treatment algorithms are largely based on practice patterns for secondary lymphedema. The mainstay of treatment focuses on complete decongestive therapy, including manual lymphatic drainage and compression therapy. For those who fail conservative treatment, surgical treatment can be an option. Microsurgical techniques have shown promise in primary lymphedema, with both lymphovenous bypass and vascularized lymph node transfers demonstrating improved clinical outcomes in a few studies.

## 1. Introduction

Primary lymphedema is an umbrella term encompassing all developmental lymphatic anomalies that result in lymphedematous swelling. Primary lymphedema is highly heterogenous in its presentation, affecting patients of any age, with lymphedema anywhere in the body ranging from mild to severe enlargement. Lymphedematous swelling may occur in isolation or in association with other syndromic features. As such, diagnosis is challenging and often missed. Unlike secondary lymphedema, the natural history of primary lymphedema is unpredictable, with some patients achieving complete spontaneous regression soon after symptomatic onset and others developing progressive disease. Indications for surgical management are, therefore, poorly defined. Treatment algorithms are largely based on practice patterns for secondary lymphedema, with very few studies providing evidence in this distinct patient population.

## 2. Incidence & Classification

The true incidence and prevalence of primary lymphedema are largely unknown due to under-reporting and under-recognition. In adults, primary lymphedema is significantly less common than secondary, comprising less than one percent of all cases of lymphedema (Goss 2019). Within the pediatric population, however, primary lymphedema is significantly more common, encompassing over 90% of cases [1,2]. Among children, symptoms of primary lymphedema appear during infancy (49.2%), childhood (9.5%) or adolescence (41%) [3]. The reported prevalence of primary lymphedema is 1.15/100,000 among individuals aged less than twenty [4], with an overall population prevalence in the United States of 1.33/1000; however, these numbers are believed to be an underestimation [5]. Epidemiological details, such as geographic regions or ethnicities of higher incidence, are lacking [6]. 

Primary lymphedema is more common amongst females, with a 3.5:1 ratio [4]; however, gender discrepancies differ based on age of presentation. Development of primary lymphedema during adolescence is more common among girls (approximately 2:1 ratio), whereas in infancy, boys are more commonly affected than girls (approximately 2:1 ratio) [1]. The significantly earlier age of onset in boys, when compared to girls, is hypothesized to be attributable to epigenetic factors that are hormonally mediated, allowing differing expressivity over the lifespan [1]. Our understanding of the relationship between sex hormone expression and lymphedema is in its infancy and is theorized based on observations of worsening lymphedema during puberty, menses, and pregnancy. Estradiol is proposed to influence the expression of VEGF-C, a growth factor implicated in lymphatic endothelial repair with a known role in lymphedema. The full extent of this relationship, and the role of sex in the development of primary lymphedema, remains to be fully elucidated and may represent an exciting area for therapeutic intervention [7]. 

Historically, primary lymphedema was classified into three categories based on the age of onset: lymphedema congenita (birth to approximately 2 years), lymphedema praecox (approximately 2–35 years), and lymphedema tarda (>35 years) [8]. Though widely used, inconsistent age ranges are reported, and the classification does not necessarily correlate with developmental age [9]. Further, this system provides no insight into pathogenesis, phenotype, or management [10]. To this end, several classification pathways have been described in which diagnosis is made based on features such as (i) syndromic associations, (ii) systemic involvement, (iii) involvement of the cutaneous and vascular systems, (iv) age of onset, (vi) other associated phenotypic findings, (vii) family history, and (viii) genetic mutations [6,11,12,13]. These pathways provide information on the natural history and clinical presentations of the many forms of primary lymphedema. 

A relationship between primary lymphedema and vascular anomalies has been proposed; nearly one-quarter of patients with primary lymphedema have concomitant vascular malformation. Unlike primary lymphedema, which is not associated with a vascular malformation in which lymphatic hypoplasia is the most common, lymphatic hyperplasia is the most frequently identified malformation amongst patients with a vascular malformation [14,15]. As such, the International Society for the Study of Vascular Anomalies (ISSVA) in 2018 included primary lymphedema in their classification system for vascular anomalies in the category of “lymphatic malformation”, in conjunction with macro- and micro-cystic lymphatic malformation [16]. Congenital lymphedema malformation lesions can be divided into truncular (lesions that develop later in development, during the formation of the lymphatic trunks, vessels, and lymph nodes) or extratruncular (lesions secondary to abnormalities in the embryonic tissues). Systemic primary lymphedema is reported as a type of truncal lymphatic thought to result from a defect of the lymphatic system that occurs late in lymphangiogenesis, whereas lymphatic malformations are attributable to defects early during development. Hereditary primary lymphedema, such as Milroy’s disease or distichiasis-lymphedema, however, is not the result of truncular defects and as such, their classification as a ‘congenital’ lymphedema has been called into question [17]. 

## 3. Differential Diagnosis

Accurate diagnosis of primary lymphedema can be challenging given the numerous other conditions that can present with limb enlargement. In a review of children referred to a lymphedema clinic, 38.7% had been misdiagnosed with primary lymphedema [18,19]. As such, the delay to accurate diagnosis of primary lymphedema is often significant, reported at more than ten years between symptomatic onset and clinical diagnosis [2]. 

The differential diagnosis for an enlarged, swollen extremity in the absence of obvious injury is wide and may include systemic causes (including cardiac, hepatic, nephrotic, thyroid diseases, hypoproteinemia), venous insufficiency or thrombosis, drug-related (rapamycin inhibitors, antipsychotics, antidepressants, anti-Parkinsonian medications, bisphosphonates), or lipedema [20,21]. Other congenital considerations include vascular anomalies (micro- or macro-cystic lymphatic malformations, capillary or venous malformations, infantile hemangiomas, and Kaposiform hemangioendothelioma), or hypertrophic syndromes (Klippel–Trénaunay, Parkes Weber, and CLOVES) [18,22]. 

## 4. Pathophysiology

Unlike secondary lymphedema, which is most commonly attributable to disease or injury to the lymphatic system, primary lymphedema arises due to intrinsic abnormalities of the lymphatics. Reduced lymphatic growth (hypoplasia, aplasia), increased lymphatic size (megalymphatic), increased number of vessels (hyperplasia), growth in the incorrect location (lymphangiodysplasia), valvular dysfunction (resulting in lymphangiectactic dilatation, lymphatic reflux, lymphorrhea), and/or functionally inadequate drainage with impaired contractility may arise secondary to genetic mutations that may be familial or arise de novo [6,8]. Results from lymphoscintigraphy findings suggest that the majority (56%) of patients have hypoplastic lymphatics (56%) or aplasia (14%) [14,15]. Furthermore, disruption in mechanisms, including initiation of lymphangiogenesis, differentiation of lymphatic structures, and mediation of cell migration and adhesion, have all been implicated in primary lymphedema [6].

Primary lymphedema may be fatal in the prenatal period or remain silent until any time in life when an imbalance between lymph production and lymph transport results in edema, chylous or non-chylous lymph accumulations and/or effusions [6]. The natural history of primary lymphedema differs significantly from secondary lymphedema. Whereas secondary lymphedema is thought to always progress, outcomes for patients with primary lymphedema are unpredictable, with progressive, stagnant, and recessive disease patterns all described. Some authors report that the age of lymphedema onset does not influence the progression and morbidity conferred by the lymphedema [18,19], whereas others suggest that patients with later onset are less likely to spontaneously improve when compared to those who developed lymphedema within the first year of life [23]. Among patients with a progressive course, there is considerable variability in the rapidity of lymphedema progression and development of lymphatic sclerosis and soft tissue fibrosis [24]. 

The pathophysiology of adult-onset primary lymphedema is not well understood. No specific germline or somatic mutations have been identified, nor has familial transmission been described. It is suggested the delayed presentation is likely attributable to less severe developmental anomalies of the lymphatic system that do not become clinically apparent until the lymphatic function fails later in life [14,15]. Lymphangiography has demonstrated that adult-onset primary lymphedema is associated with less severe aplasia or hypoplasia compared to early-onset disease; on lymphoscintigraphy, patients with adult-onset primary lymphedema have significantly more dermal backflow (73%) compared to children (31%) [14,15]. With aging, contraction frequency decreases, impairing pumping activity and slowing the velocity of lymphatic flow. In both animal models and human lymphoscintigraphy, lymphatic drainage is shown to slow with age [20]. 

### 4.1. Genetics

Primary lymphedema is most commonly idiopathic, as approximately 70% of patients with primary lymphedema have no identifiable underlying genetic defect; however, it is likely the genetic basis of disease in these patients simply has not yet been discovered [6]. Genetic predisposition to primary lymphedema is most commonly attributable to an inherited autosomal dominant mutation with variable penetrance [21]. Even within a single family, significant differences in lymphedema phenotype are common. Over thirty genes and loci have been identified and implicated in lymphangiodysplasia or lymphangiogenesis [6,24,25]. Identified genes implicated in primary lymphedema include *FLT-4*, *VEGFC*, *FOXC2*, *GJA1*, *PTPN14*, *SOX18*, *HGF*, *GJC2*, *GATA2*, *FAT4*, *CCBE1*, *ADAMTS3*, and *NEMO* [1]. Specific molecular pathways implicated in the disease process include the PI3K/AKT, VEGF-C/VEGFR, RAS/MAPK, and HGF/MET pathways. The details of these mutations, their molecular pathways, and their clinical manifestations have been thoroughly described [6,26]. A full review of genetic mutations associated with syndromic lymphedema is reviewed by Pateva and Brouillard and is beyond the scope of this article [6,21]. 

Genetic testing is indicated when patient history and physical exams are suggestive of an underlying syndrome. Current methods of genetic testing using blood or saliva are reported to have low efficacy in accurately diagnosing primary lymphedema [26]. These limitations are attributable to insufficient knowledge of the genetic basis of lymphedema resulting in a paucity of targets to specifically test. In this context, whole exome sequencing, when available, is becoming an increasingly popular option. In addition to its diagnostic functions, routine genetic analysis academically provides greater insight into genotypic variability and genotype-phenotype correlations [6]. 

Lymphedema is one of the multiple clinical features characteristic of numerous genetic syndromes, including Hennekam’s, Aagenaes’, microcephaly-chorioretinopathy-lymphedema, Mucke’s, Noonan syndrome, Turner’s, Prader–Willi, CHARGE, Irons–Bianchi, Emberger, oculo-dento-digital, Phelan–McDermid, and yellow nail syndrome [11,27]. We herein include a discussion on the most common genetic syndromes associated with lymphedema. 

### 4.2. Milroy Disease

Milroy’s disease was first described by Dr. Milroy in 1892 as an inherited, congenital, nonprogressive lymphedema of the lower extremity [27,28]. Milroy’s disease is the most common hereditary form of primary lymphedema [29]. This syndrome is attributable to an autosomal dominant mutation of gene *FLT4* on gene 5q35.3 which encodes for vascular endothelial growth factor receptor-3 protein (VEGFR-3). Penetrance is relatively low; approximately 50% of patients with the mutation do not have clinically detectable lymphedema [6]. This mutation disrupts the tyrosine kinase domain of VEGFR, which in rodents is reported to disrupt the growth and development of the lymphatic endothelial cells [29,30]. *Vegfr3* heterozygous knockout mice develop a lymphedema phenotype secondary to aplastic lymphedema [31]. By contrast, in humans, Milroy’s disease is not associated with lymphatic aplasia but rather a functional failure: significant impairment of initial lymphatic absorption hypothesized to be attributable to poor endothelial junction flap valves and poor lymph transport associated with vessel hypoplasia [27,31]. These patients additionally have large superficial veins that have a propensity for reflux and valve failure [27]. Patients with Milroy disease present perinatally with bilateral lower extremity lymphedema, often associated with “woody” overlying skin and prominent veins. Lymphedema is typically confined to the feet and ankles, with slanting ‘ski-jump’ toenails due to the disease of the nail bed [11]. Diagnosis can be suspected as early as 12 weeks of gestation with ultrasound identification of pedal edema [27]. 

The term “Milroy disease” has erroneously been used as a ‘catch-all’ phrase to encompass all infants presenting with lymphedema present at birth or within the first year of life. The correct terminology refers to a familial form of primary lymphedema characterized by edemlower extremity edemasent at birth. Historically, patients required both a consistent phenotype and a positive family history to be diagnosed with Milroy’s; however, de novo mutations can occur in patients with no family history. Therefore, diagnostic criteria of Milroy’s disease now include infants diagnosed with lower extremity lymphedema with either a positive family history and/or documented *FLT4* mutation [18,19,27]. Disease phenotypically consistent with Milroy’s but without a positive family history of *FLT4* mutation should be termed “Milroy-like lymphedema” [11]. Based on these criteria, patients with involvement extending beyond the lower extremity, a delayed presentation, or a negative for a VEGFR-3 mutation should not be included in this diagnosis. Of note, not all patients presenting with a phenotype consistent with Milroy’s disease have a VEGFR3 disruption; one study identified that among these patients, the mutation was present in 72% with a positive family history and 64% without [27]. 

### 4.3. Meige Disease

First described in 1898, Meige disease was initially identified as familial lymphedema presenting in the lower extremity with onset during adolescence [32]. Over a century later, no genetic etiology underlying Meige disease has been identified; therefore, current recommendations suggest that a positive family history is necessary for this diagnosis [18,19]. The lymphatic system in these patients is reported to function at approximately 10% normal capacity [5]. Lymphedema is typically limited to below the knee, and Meige disease has no other associated features [11]. 

### 4.4. Distichiasis-Lymphedema

Distichiasis-lymphedema syndrome is a single-gene disorder with high penetrance and variable expressivity [27]. Over thirty *FOXC2* mutations have been described as affiliated with distichiasis-lymphedema syndrome, and phenotypic expressivity is more common amongst female mutation carriers than males [33]. Mutation of the *FOXC2* gene is associated with lymphedema in combination with distichiasis, defined as the presence of a second row of eyelashes emerging from the Meibomian glands. The lymphatic abnormality associated with this autosomal-dominant syndrome is typically hyperplasia [33] though others suggest this syndrome is more likely attributable to lymphatic and venous valve failure, a theory corroborated by lymphoscintigraphy and venous duplex ultrasound analysis [27]. Lymphedema presents in the bilateral lower limbs typically in adolescence, though manifestation at birth or in the fifth decade of life has also been described [10]. Distichiasis-lymphedema may be associated with congenital heart disease, pterygium, lid ptosis, cleft lip/palates, and/or venous malformations [33]. 

### 4.5. Turner Syndrome 

Over 40% of children with Turner’s will have lymphedema at birth, most commonly affecting all four limbs distally [22]. Cervical lymphedema can contribute to a webbed neck and a low posterior hairline. Lymphedema in this patient population typically disappears by 2–3 years of age, although some patients see a recurrence in at least one limb at some point in their lives [23,34]. The lymphatic abnormality in Turner Syndrome is proposed to be due to a failure of lymphatic absorption at the distal level [34]. 

### 4.6. Noonan Syndrome

Noonan syndrome is an autosomal dominant genetic disorder commonly associated with the mutation of *PTPN11*. The syndrome is characterized by multiple congenital anomalies, including short stature, webbed neck, facial abnormalities, intellectual disabilities, and heart defects [33]. The development of lymphedema in association with Noonan syndrome is inconsistent [23]. Lymphangiectasia of the gastrointestinal tract and chylothorax has been described [5]. 

### 4.7. Hennekam Syndrome

Hennekam syndrome is due to mutation of *CCBE1* (collagen and calcium binding EGF-domain 1) resulting in systemic marked lymphatic dysplasia. [5]. This syndrome is characterized by lower limb lymphedema and lymphangiectasia, facial anomalies (flat face, broad nasal bridge, and hypertelorism), and varying severities of mental retardation [11,33]. Lymphangiectasia may develop in the gastrointestinal tract, lungs, pleura, pericardium, thyroid, or kidneys [5]. Associated features may include hypothyroidism, glaucoma, seizures, hearing loss, and renal anomalies [11]. Patients typically have symptomatic improvements in their first year of life; then, the disease gradually progresses over time [5]. 

## 5. Clinical Presentation 

Primary lymphedema is predominately a clinical diagnosis. A detailed history and physical examination alone can accurately identify lymphedema in approximately 90% of patients [18,19], and should be performed in all infants, children, or adolescents with a swollen extremity to establish an accurate diagnosis [23]. Specific points to elucidate about the swelling include the age of onset and progression of edema, laterality, aggravating and alleviating factors, infections, and associated discomfort and/or pain. Screening for etiologies, including a complete past medical, family, travel, and trauma history. A complete perinatal history is indicated for pediatric patients [2]. A physical exam should elucidate the location and extent of lymphedema, identify tissue fibrosis and pitting, and evaluate for soft tissue infection [23]. Patients should be screened for signs and symptoms of external compression of the lymphatics by a mass, systemic illness, or known associated syndromes. 

Primary lymphedema most commonly involves the lower extremity, initially affecting the foot and progressing proximally (Figure 1). The dorsum of the foot has a ‘buffalo hump’ secondary to swelling, with a positive Stemmer sign, in which the skin on the dorsum of the toe cannot be pinched between the examiner’s thumb and index [10,23] (Figure 2). Stemmer sign is reported to be 92% sensitive and 57% specific for lymphedema [1]. Edema effaces the natural transverse creases of the foot and fills the retromalleolar spaces, resulting in a loss of definition of the foot and ankle [23]. Patients often describe a sense of heaviness which may result in functional dysfunction, specifically the inability to perform normal age-based physical activities such as crawling or ambulating (lower extremity), feeding or writing (upper extremity), playing sports, or engaging in social activities. Acute worsening of lymphedema can cause pain [10]. The extent of pain amongst these patients is largely under-investigated, but one study suggested that one-in-four children experience significant pain, and one in five have pain resulting in impairment in daily activities [1]. Care must be taken to rule out other causes prior to attributing pain to lymphedema. 

Age of onset is a significant predictor of the extent of disease and clinical progression, with an earlier age associated with more severe disease. Less extensive disease may remain subclinical and not present until later in life, when influences such as hormones, injury, often minor, or inflammation cause an insult to a weakened lymphatic system. Younger patients are more likely to have bilateral disease, involvement of the upper extremity and/or genitalia, and systemic lymphatic dysfunction. Involvement of the upper extremity is reported in almost 10% of patients and most commonly presents in patients in infancy [14,15]. Cutaneous involvement, particularly involving the toes, is most common in children with earlier onset [23]. Skin changes associated with primary lymphedema may include hyperkeratosis, hyperpigmentation, papillomatous/verrucous lesions, and lymphorrhea [3,10]. 

History and physical exams must include a complete review of systems to rule out systemic involvement. Lymphangiectasia, or the dilation of lymphatics, results in lymphatic leakage into nearby tissue, causing pathology in the cardiac (pericardial effusion), pulmonary (chylous pleural effusion), gastrointestinal (protein-losing enteropathy, malabsorption) thyroid, and renal systems [5,10]. An earlier age of onset is associated with a higher risk of systemic involvement. Among infants with onset within the first year of life, the risk of lymphatic comorbidities, including chylothorax, chyloabdomen, and chyluria, is as high as 75% [24]. A review of systems and a complete physical exam should additionally include screening for stigmata of common syndromes associated with primary lymphedema, as outlined above. Signs that should prompt further evaluation for syndromic involvement include yellowed nails, profuse warts, distichiasis, vascular malformations, hypertrophy and/or asymmetry in limb length, facial dysmorphism, and/or intellectual retardation [2]. Hematologic evaluation, including a complete blood count, renal and hepatic function, thyroid function, and albumin levels, should be assessed in all children with lymphedema [1].

A prenatal diagnosis of primary lymphedema based on ultrasonographic findings is reported [2,29]. Ultrasound findings may include anasarca, serous effusions, increased volumes in the hands or feet, or nuchal translucency [2]. Prenatal suspicion for lymphedema should be further evaluated with amniocentesis and evaluation of the chorionic villi to assess for genetic mutations and indications of metabolic diseases and/or lysosomal stage disease [5]. 

Cellulitis is less common in children with lymphedema than in adults [22]. Wounds, intertrigo between toes, and ingrown nails serve as common entry points for bacteria. A cohort study by Quéré reported the incidence of cellulitis in children with primary lymphedema as 4.2 episodes per 100 person-years, with a mean time to infection of 5.5 years following lymphedema diagnosis. Importantly, 45% of affected patients required hospitalization at least once for sepsis [35]. 

## 6. Imaging 

When the history and physical exam are equivocal for diagnosing lymphedema, imaging modalities can be used to confirm the diagnosis. Lymphoscintigraphy has a 92% sensitivity and 100% specificity for identifying lymphedema and is considered the gold standard for diagnosis [10]. Drainage patterns, dermal backflow, interruptions in lymphatic channels, and the presence of regional lymph nodes can all be assessed using this modality, including the assessment of both the superficial and deep lymphatics [5]. As lymphoscintigraphy requires patient cooperation, it is typically performed after the age of 7–8 years [2]. Lymphoscintigraphy remains non-standardized with respect to the type of radiotracer and dose of radioactivity, the volume, number and site of injections, the protocolized physical activity required, and the imaging time [10]. Spatial resolution can be improved by combining scintigraphy with SPECT-CT; the presence of dermal backflow, the extent of the lymphatic disorder, and the presence of lymph nodes can all be further elucidated by combining these two techniques [6,36]. Lymphoscintigraphy is not limited by tissue depth and may provide more information for obese patients, whereas other modalities may have difficulty penetrating deeply enough. Findings consistent with lymphedema include delayed transit time, dermal backflow, asymmetry in nodal uptake, presence of collateral lymphatic channels, and tracer uptake in the deep lymph nodes [6]. 

Indocyanine green (ICG) fluorescent angiography has become increasingly popular for diagnosing lymphedema and delineating the extent of the disease as well as for preoperative planning. This technique is less invasive and less costly than lymphoscintigraphy. It has a high sensitivity and negative predictive value for accurate diagnosis, with a similar diagnostic ability to evaluate disease severity as lymphoscintigraphy [10]. ICG is well-reported to be safe for pediatric patients [37]. The primary disadvantage of ICG angiography is that it allows for visualization of only the superficial lymphatics located less than 2 cm deep in the subcutaneous tissue [10]. The specific pattern of dermal backflow and enhancement on lymphangiography in primary lymphedema can be classified into four types: (i) distal dermal backflow (DDB, dermal backflow identified distally but not proximally in the groin or axilla), (ii) proximal dermal backflow (PDB, dermal backflow identified from groin to foot or axilla to hand), (iii) less enhancement (LE, linear pattern only in the distal limb, no dermal backflow), and (iv) no enhancement (NE, no enhancement). Whereas dermal backflow patterns reflect the degree of lymphatic obstruction, loss of enhancement indicates severe hypoplasia and/or aplasia. This classification scheme provides insight into the etiology of primary lymphedema and guides management [38]. For example, lymphography evidence of obstructive patterns (PDB, DDB pattern) may suggest surgical management to bypass the obstruction (lymphaticovenous bypass) versus evidence of aplasia (NE pattern) may merit a nodal transfer or debulking [38]. 

MR lymphangiography (MRL) is a relatively new technique in which intracutaneous injection (gadolinium contrast, ferumoxytol) and intravenous injection (iron particles, gadolinium-based agent, respectively) allow static visualization of the lymphatic vessels and dynamic monitoring of lymph flow [6,21,39]. Noninvasive MRL has also been described without contrast. This imaging modality provides differentiation between hypotrophic and hypertrophic lymphedema patterns, an essential distinction for surgical management [39]. The availability and reproducibility of high-quality images significantly deter the widespread adoption of MRL imaging for primary lymphedema. 

Other imaging modalities that evaluate primary lymphedema aim to rule out other causes of limb edema. CT or MR imaging of the axilla and/or inguinal region is indicated in adult patients to rule out mass lesions, collections, or thrombosis. Venous duplex ultrasound is commonly used to rule out vascular lesions and venous thromboses [10]. 

Ultra-high frequency ultrasound is showing some promising results in imaging lymphatic channels in real-time, including contractility.

## 7. Treatment

The goals of treatment are to prevent progression, restore limb function and cosmesis, avert complications of lymphedema such as cellulitis, and improve quality of life [2,10]. Most treatment recommendations for primary lymphedema are based on clinical practice guidelines established for secondary lymphedema. A greater understanding of the natural history of various genetic mutations and syndromes will allow for the individualization of treatment modalities [6]. Current management techniques are summarized in Figure 3. 

Management of primary lymphedema requires an extensive multidisciplinary team which must include certified lymphedema therapists (physiotherapy, occupational therapy), a team of physicians (pediatrician, surgeon, internist, dermatologist, bariatric surgeons), psychologists, dieticians, social workers, podiatrists, and orthopedists [2]. The most important person on the team, however, is the patient and their family, as their active, ongoing engagement in care is essential to success. It is imperative that patients understand the natural history of their disease, including the potential complication and the need for lifelong therapy and self-management strategies. This patient population is at an increased risk of depression and/or anxiety; therefore, open dialogue with regular check-ins to address patient concerns and questions is important for monitoring patient well-being [40,41].

Complete decongestant therapy (CDT) is the mainstay of treatment for primary lymphedema and includes manual lymphatic drainage, compression, exercise and weight loss, and skin care. CDT is divided into two phases: phase one aims to reduce limb volume and improve skin quality, and phase two is the maintenance period to prevent progression [10]. Despite this being the first-line treatment recommended in this patient population, there remains a paucity of evidence supporting or refuting the use of CDT in primary lymphedema [8]. A recent retrospective review of patients with primary lymphedema assessed during CDT with lymphoscintigraphy found that successful limb reduction was associated with older age, with a 0.16% volume reduction for each additional year of age. Other predictors of greater volume reduction with CDT were patients with a BMI > 40, a positive history of cellulitis, and lymphoscintigraphy positive for dermal backflow [42,43]. 

Physical activity should be encouraged in children with primary lymphedema for its physiologic benefits and to manage patient weight. Exercising the lymphedematous limb stimulates the muscle to propel venous return and lymphatic circulation centrally [21]. Additionally, through deep inspiration, a negative intrathoracic pressure is generated, which further propels lymphatic fluid proximally [10]. Patient BMI is an important predictor of disease progression, with obesity conferring an increased risk of disease progression and greater morbidity [21]. 

Skincare is integral to the prevention of skin desiccation and breakdown and to prevent cellulitis. The skin should be regularly evaluated with specific foot exams for ingrown toenails, heel fissures, onychomycosis, and inter-toe webspace intertrigo and desquamation [2]. These patients are at a higher risk of cutaneous fungal infections, with tinea pedis reported in 43% of patients [1]. Cellulitis should be treated promptly with antibiotics—oral therapy is typically first-line; however, intravenous treatment may be required in the setting of sepsis [2] or in the absence of an adequate response to oral antibiotics within 24 h. Prophylactic low-dose antibiotics are recommended for patients who develop three or more episodes of cellulitis within a one-year timeframe [18,19]. 

Manual lymphatic drainage (MLD), as in secondary lymphedema, refers to gentle massage of the lymphedematous tissue, typically starting centrally at the abdomen and then working distally. Avoidance of deep tissue massage is important to prevent injury to the lymphatic structures [21]. Parents can provide massage therapy to their children, as instructed under the supervision of trained lymphedema therapists, to increase the child’s comfort with the therapy. As the child ages, they should be included in the therapy to encourage autotomy [2]. 

Compression therapy is the final element of CDT. Graded compression from distal to proximal using short-stretch bandages prevents reaccumulation of fluid. Compression is typically a lifelong commitment, including following surgical treatment. Regular use of compression garments is associated with lower levels of pain or numbness and improved range of motion [44]. Compression garments require frequent re-sizing during childhood to accommodate for normal growth. The use of compressive dressings for babies and infants has not yet been established [2].

### 7.1. Surgical Treatment

Specific indications for surgical interventions in primary lymphedema remain controversial, but it is generally accepted that operative management should be reserved for patients with progressive disease and significant morbidity despite compliance with all conservative measures [18,19,21]. The goals of surgical intervention are therapeutic and non-curative; it is imperative that patients understand that postoperatively lifelong compression with still be required. As with secondary lymphedema, surgical options can be sub-classified as physiologic or debulking.

Evaluation of postoperative outcomes is challenging in pediatric patients due to the need to correct measurements for normal growth [24]. When the disease is unilateral, measurements from the contralateral limb can provide normalized data against which the affected limb can be compared. Differentiation between expected growth and disease progression, however, can be challenging for children with bilateral limb involvement; therefore, regular follow-up with well-documented consistent measuring is required to assess trends. 

### 7.2. Debulking Surgery

Debulking surgical options are liposuction and/or surgical excision. Liposuction is the first line of debulking surgery for symptomatic patients. Indications for liposuction, however, are largely unknown. Liposuction is generally not recommended for pitting edema, as the aim is to remove fat, not fluid [10]; however, others argue that once the tissues become fibrosclerotic and non-pitting edema develops, fat may no longer be available for liposuction [10,45]. Suction lipectomy is reported to reduce excess limb volume by over 100% in the upper extremity and 75% in the lower extremity [18,19], and results have been shown to be sustained for over 20 years. Postoperatively, patients require lifelong around-the-clock compression to maintain the benefits of liposuction [6].

Skin excision should be reserved for patients with involvement of the genitalia or in the presence of severe disease of the extremities [18,19]. Severe psychological morbidity is secondary to the appearance of the enlarged limb is also described as a reasonable indication [10]. Others argue that excision should not be performed on the extremities and they should be reserved for scrotal edema where other treatment options are not possible [6]. Major excisions portend significant morbidity perioperatively (blood loss, graft failure, wound healing problems) and in the long term (scarring, graft contracture) [10]. Worsening distal edema, usually of the dorsal foot, is usually a side effect of circumferential excision and grafting.

### 7.3. Physiologic Surgery/Microsurgery

Two forms of physiologic management are the lymphaticovenous bypass or anastomosis (LVB/LVA, Figure 4) and the vascularized lymph node transfer (VLNT) (Figure 5). Some authors have suggested that lymphatic vessel hypoplasia is not amenable to these physiologic procedures [14,15]. The use of microsurgical techniques for the management of primary lymphedema, particularly in a pediatric population, is in its infancy, and significantly more work is required to fully understand the indications, patient selection and anticipated outcomes of these procedures; the available evidence is limited to observational non-randomized studies. Meta-analysis of the available literature is precluded by the considerable variability in outcomes and inconsistency in the reporting of the degree of postoperative improvement [4]. Due to these limitations, it has been suggested that lymphatic reconstruction is not indicated for the management of primary lymphedema [2]. Early studies, however, conclude that microsurgical lymphatic reconstruction with LVB and VLNT is safe, reliable, and effective in decreasing limb volume, reducing rates of cellulitis, and improving quality of life [8,46]. 

Indications for LVB or VLNT are largely based on imaging findings. It is generally agreed that lymphatic hyperplasia or obstruction can be treated with LVBs, and severe lymphatic hypoplasia or aplasia requires VLNT. Specific imaging findings suggestive of these pathologic processes, however, remain contestable. Some authors suggest that lymphatic channel obstruction includes proximal or distal dermal backflow patterns, and hypoplasia or aplasia are indicated by non-enhancement on angiography [10,38]. Others, however, challenge this recommendation and suggest a distal dermal backflow pattern is more indicative of hypoplasia and should suggest VLNT instead of LVB [24], a conclusion supported by the secondary lymphedema literature in which distal dermal backflow is reported to suggest pre-existing lymphatic hypoplasia, and LVBs are not recommended [47]. 

Cheng et al. (2020) expanded the indications for physiologic surgeries beyond lymphangiogram findings. In addition to lymphangiography demonstrating patent lymphatic ducts and/or partially obstructed ducts, other indications for LVB included grade 0–2 disease, a limb circumference difference of less than 20%, and less than five years since symptom onset. Beyond the absence of patent ducts, they further suggest that VLNT is indicated with a circumference difference greater than 20%, and prolonged symptoms [8]. It has further been suggested that LVBs are contraindicated in the presence of concurrent venous flow impairment such as varices, venous hypertension, and/or valvular incompetence [25]. 

The age at which physiologic surgery should be performed remains contentious. Some authors suggest that, like secondary lymphedema, LVBs should be performed for early-stage disease [20]. It is rationalized that by allowing the disease to progress, progressive lymphatic damage may impede surgical effectiveness [10]. However, others argue that early surgical intervention may not be necessary or even possible in primary lymphedema patients. As discussed above, the natural history of primary lymphedema differs significantly from secondary; approximately half of the the patients in the two studies were found to have no worsening of swelling or functional deficits throughout up to 27 years of follow-up [18,19,48]. It also must be noted that unlike secondary lymphedema where patients can be monitored clinically and with angiography to identify disease early in its clinical course, patients with primary lymphedema often are not diagnosed until symptomatic, precluding earlier/pre-clinical intervention. A consensus document from the Union of Phlebology 2013 specifically recognized the controversy regarding the optimal timing for surgery and that surgery performed more than one year following diagnosis increases the risk of failure. They do not necessarily advocate for early intervention but rather a shorter waiting period for surgery once deemed appropriate. They recommend that patients who have failed conservative measures or continue to progress in disease should be advanced to surgical intervention without delay to increase the chance of success [45]. 

### 7.4. VLNT

Vascularized lymph node transfers are commonly cited in the primary lymphedema literature as a treatment option for aplastic or severely hypoplastic lymphatic disease or in the presence of severe lymphosclerosis; however, these conclusions are based on very few studies. It has been suggested that lymph node transplantation may induce lymphangiogenesis [39]. A retrospective review of fifteen patients who underwent VLNT for primary lymphedema found patients had an average of 3.7 cm of limb circumference reduction as well as a statistically significant (i) decrease in the number of episodes of cellulitis and (ii) improvement in their quality of life [46]. Similarly, Ciudad et al. reported on eleven patients treated with VLNT who had an average circumference reduction of 19%; of note, this was significantly less improvement than patients with secondary lymphedema who had a 25% circumference reduction [49]. Others have reported that following VLNT, complete normalization of limb volume is achievable in 20% of patients [6]. Patients with primary lymphedema who undergo VLNT may be at an increased risk of developing heparin-induced thrombocytopenia and thrombosis (HITT), which increases the risk of flap complications [50]. Case series have suggested that combining VLNT with lipectomy may improve limb circumference and patient satisfaction [30]. 

As with secondary lymphedema, possible donor sites for VLNT include submental, groin, axilla, supraclavicular, omental and jejunal mesentery, and the choice is based on minimizing the risk of iatrogenic injury [51]. Choosing a safe donor site, however, is a particular challenge in primary lymphedema, given the risk of sub-clinical lymphatic dysfunction at the donor site, which may increase the risk of postoperative iatrogenic donor-site lymphedema [14,15]. The most common donor sites reported for primary lymphedema in the literature to date are submental (83%) [4]. The recipient site depends on the extent of lymphedematous disease; the VLNT should be placed at the level of the knee for isolated distal disease, whereas placement in the inguinal region is indicated for whole-lower limb involvement. Placement of the VLNT further away from the site of maximal swelling requires some lymphatic channel function. The severe lymphedematous disease may require more than one flap, one placed proximally and one more distally [39]. Complete reconstruction of a VLNT with efferent lymphaticolymphatic anastomosis is reported to successfully treat primary lymphedema [52]. A summary of reported VLNTs in primary lymphedema is summarized in Table 1.

### 7.5. LVB

Several studies have investigated the use of lymphaticovenous bypasses (LVBs) for the treatment of primary lymphedema. Historically, it was controversial whether LVBs would be beneficial, given prior reports that 80% of patients with primary lymphedema had aplastic or severely hypoplastic lymphatic channels, and therefore most of the patient population would not benefit [55]. Other authors, however, have countered that though the morphology of the lymphatic system is more variable compared to secondary lymphedema, over 80% of patients with primary lymphedema have a lymphatic channel greater than 0.3mm and are thus suitable candidates for LVB [56]. Additionally, because primary lymphedema globally damages lymphatic drainage routes, multiple LVB sites or repeated surgical LVB procedures may be required to see clinical improvement [57]. Further large-population studies are required to fully investigate this important question; however, to date, initial reports of LVBs in the literature have been promising, particularly among patients that developed lymphedema at adolescence or beyond. An overview of reported LVB studies in primary lymphedema is summarized in Table 2.

Hara et al. reviewed 79 lower limbs with primary lymphedema treated with LVB. They reported that only children with onset after age eleven had a significant improvement in leg circumference [24]. They conclude that while LVBs are not contraindicated for younger children, outcomes are less predictable. Lymphangiography may help ascertain which patients are more likely to benefit, as authors noted patients with no backflow on ICG angiography had greater improvements following LVB; by contrast, only half of the patients with isolated distal backflow demonstrated any effect following LVB, and the improvement was minimal. Similarly, Yoshida recently reported outcomes of 150 lower limbs with primary lymphedema treated with LVB [20]. In keeping with Hara’s results, authors found that patients with an age of onset older than 65 years had greater improvements following LVB when compared to younger age groups. They additionally noted that older patients had higher rates of lymphatic detection, a larger vessel diameter, a decreased interval between lymphedema onset and LVA surgery, and a greater number of LVBs performed. All these factors likely contributed to improved outcomes.

Outcomes of LVB appear to differ in primary lymphedema compared to secondary. Drobot et al. reported a 37% volume reduction at 6 months postoperatively among lower extremities treated with LVBs, which was significantly less than the 55% reported in patients with secondary lymphedema [59]. A less significant difference was identified by Demirtas et al., who reported a 56% mean percent volume decrease one year following LVB among patients with primary lymphedema compared to a 60% reduction identified in patients with secondary lymphedema [56]. This begs the question of the importance of patient selection. The lymphatics of the former group were noted to be smaller with variable morphology between channels even within the same incision, suggesting not all channels are affected to the same degree and searching for viable channels may be fruitful even in the presence of pathology.

Cheng et al. compared LVB and VLNT for primary lymphedema in two studies. The type of surgery was based on ICG angiography identifying patent lymphatics (LVB) or non-patent lymphatics (VLNT). Though both LVB and VLNTs were reported to reduce the number of episodes of cellulitis and decrease limb circumference, VLNTs had greater improvements in volumetric reduction and improved quality of life [46]. Additionally, they noted that 84.6% of patients in the study required VLNT due to the severity of the disease at the time of presentation. Both studies, however, are based on a very small number of patients (<20) which were performed in different patient populations; therefore, it is difficult to draw clinically relevant conclusions.

## 8. Conclusions

Primary lymphedema is a complex and heterogeneous group of lymphatic disorders that arises from the intrinsic disease within the lymphatic system and comprises a large number of different pathophysiologies. Primary lymphedema differs from secondary lymphedema in terms of an unpredictable disease course, often progressing more slowly than secondary lymphedema and only clinically presenting later in the disease. The mainstay of treatment focuses on CDT, although evidence in the primary lymphedema population is scant. For those who fail conservative treatment, surgical treatment can be an option. Both LVB and VLNT have shown efficacy in primary lymphedema, although larger and longer-term studies are required.

## Figures and Tables

**Figure 1 medicina-59-00894-f001:**
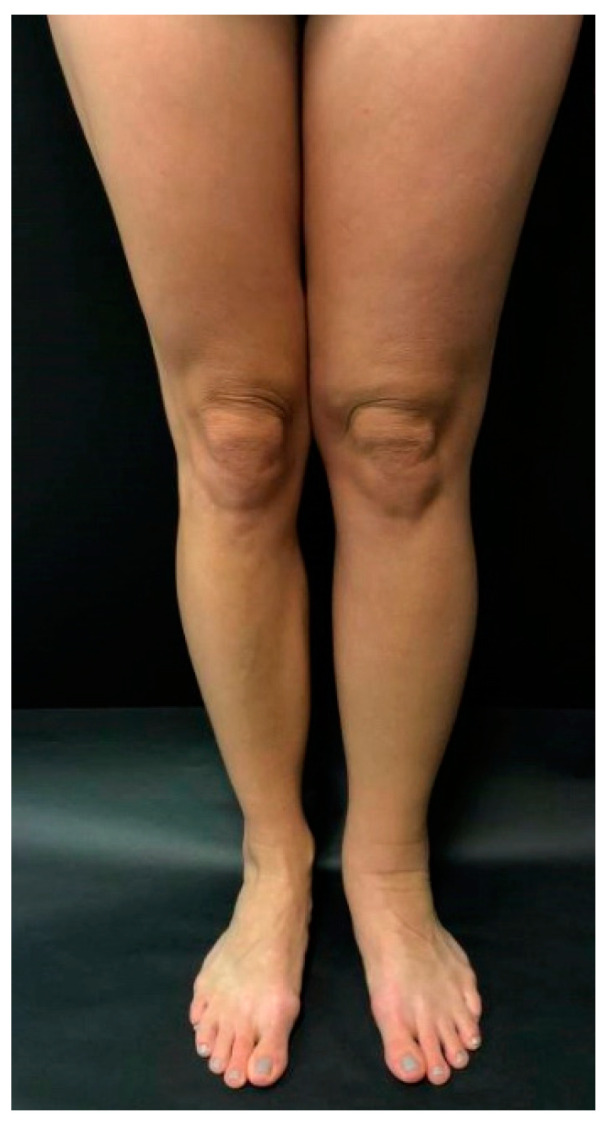
A patient with primary left lower extremity lymphedema that developed during adolescence.

**Figure 2 medicina-59-00894-f002:**
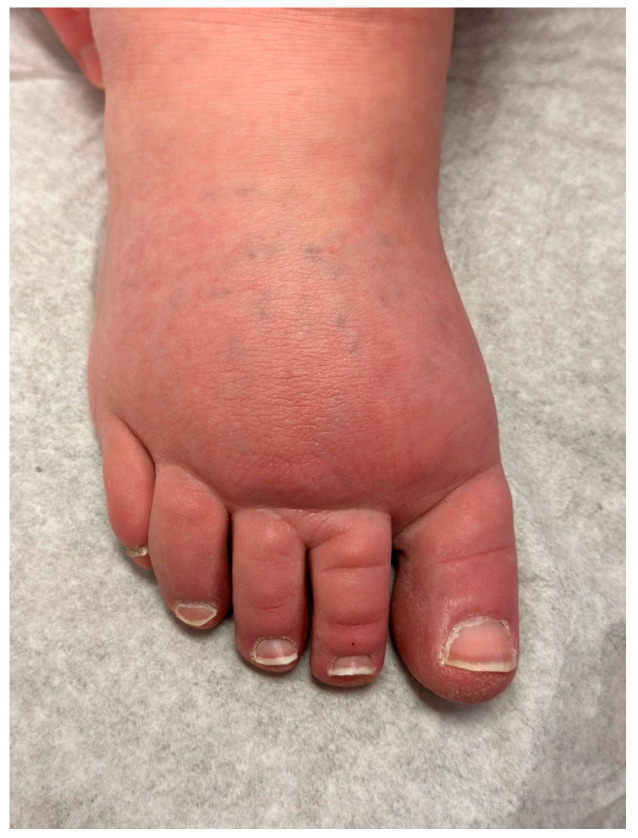
Typical dorsal foot lymphedema. Accumulation of lymph with tissue overgrowth of the dorsal foot eliminates natural creases and precludes the pinching of the dorsal foot skin between the examiner’s fingertips (“stemmer sign”).

**Figure 3 medicina-59-00894-f003:**
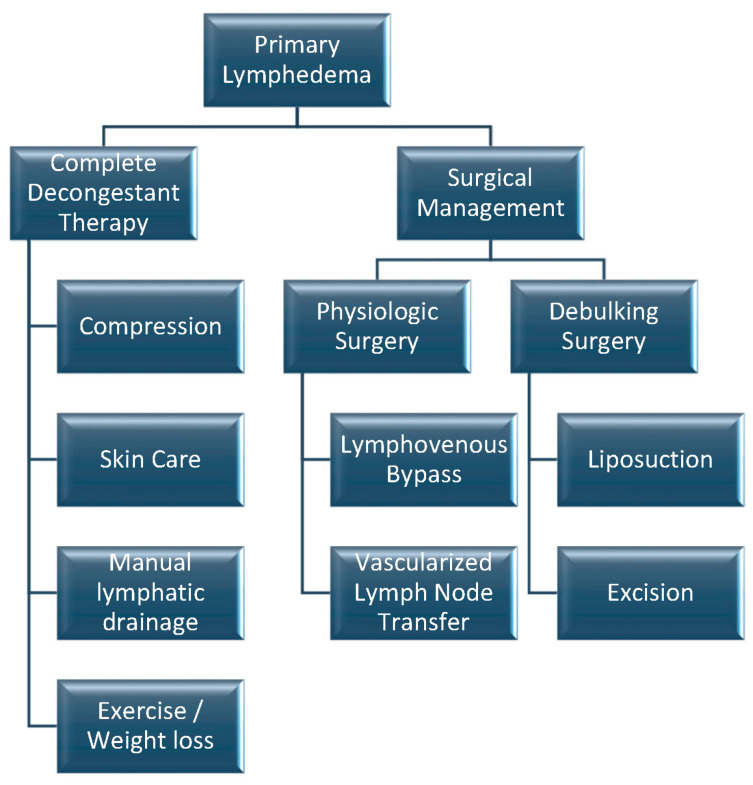
Management algorithm for primary lymphedema.

**Figure 4 medicina-59-00894-f004:**
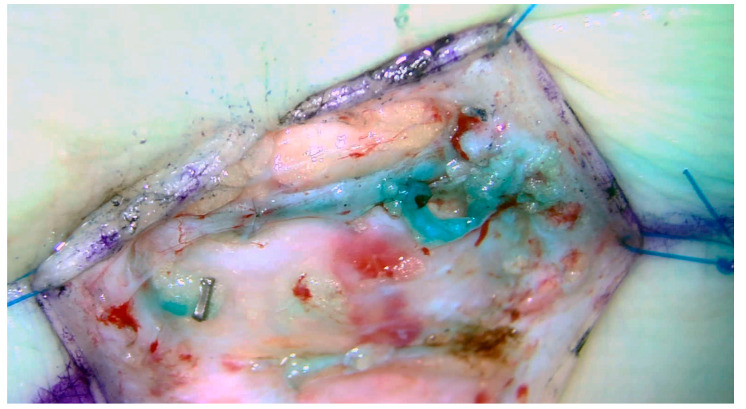
Lymphaticovenous bypass. End-to-side anastomosis between the lymphatic channel and a local vein used as a physiologic treatment for obstructive primary lymphedema.

**Figure 5 medicina-59-00894-f005:**
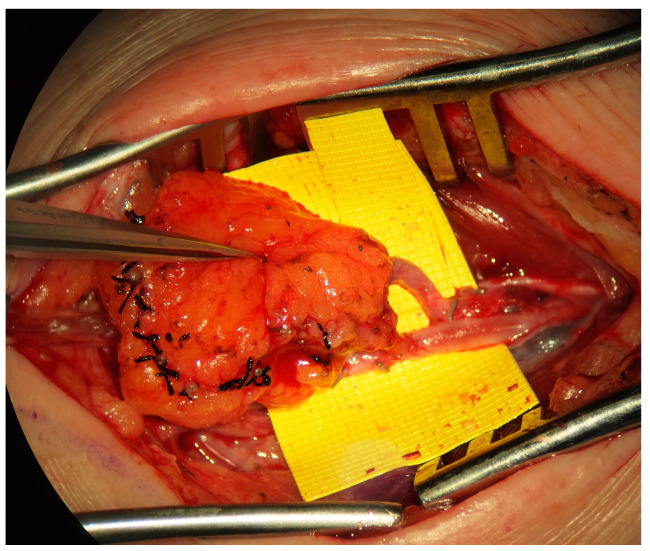
Vascularized lymph node transfer. VLNT harvested from the jejunal mesentery based on branches from the superior mesenteric artery used as a physiologic treatment for aplastic primary lymphedema.

**Table 1 medicina-59-00894-t001:** Reported vascularized lymph node transfers for primary lymphedema.

Authors	# Limbs (Average Age)	Symptom Duration (Months)	Donor Site	Recipient Site	Outcomes	Time to f/u
Cheng 2018 [46]	15 (30 years)	24.1 ± 5.7	Submental	Most dependent region of extremity	Circumference reduction (3.8 ± 3.0 cm above knee, 4.0 ± 2.5 cm above ankle) Decreased cellulitis Improved quality of life	18 months
Ciudad 2020 [49]	11 (range 26–53 years)	NR	NR	Distal ankle/wrist	Circumference reduction (19%)	24 months
Cheng 2020 [8]	11 (8.9 years)	105.8	Omentum (1), submandibular (10)	1 proximal, 10 distal	Circumference difference between limbs (from 28.3% preoperative to 21.1% postoperative)	39 months
Venkatramani [53]	1 (13 years)	11 years	IL axilla	Anterior tibia	Circumference reduction (9%) Improved EQ5D health state	6 months
Chen 2019 [54]	6 (30 years)	NR	Omentum	Ankle	Circumference reduction (4.2 ± 2.2 cm) Decreased cellulitis Improved quality of life	10.7 months
Yamamoto 2016 [52] *	1 (49 years)	5 years	Lateral thorax	Iliac	No cellulitis Lower extremity lymphedema index (264)	18 months

* This patient had VLNT with efferent lymphaticolymphatic anastomosis. NR—not reported. #— “number of”.

**Table 2 medicina-59-00894-t002:** Reported lymphaticovenous bypasses for primary lymphedema.

Author	# Limbs (Age Years)	Symptom Duration	Surgical Details	Outcome	Average Time to f/u
Cheng 2020 [8]	2 (13 & 10)	100 & 60 months	ETE/ETS	Circumferential difference between limbs (7.7% preop, 2.2% postop)	63 & 31 months
Demirtas [56]	80 (32.5 ± 15.6)	8.1 ± 6.9 months	2.3 ± 0.5 anastomoses per limb	Volume reduction (56.2% ± 22.8%) LVB possible in 83.8%	13.3 months
Hara 2015 [24]	79 (42 years)	10.6 years	4.5 anastomoses per limb	Circumference reduction in patients >11 years of age	610.7 days
Cheng 2018 [46]	4 (37 ± 20 years)	24.6 ± 1.3 months	1 LVA/limb ETE or ETS	Circumference reduction (1.3 ± 2.0 cm above knee 1.5 ± 4.4 cm below knee)	26 months
Yoshida 2020 [58]	136 (73.6 ± 11.8 years)	6.1 ± 7.2 years	3.8 ± 1.5 (bilateral) vs. 2.2 ± 1.6 (unilateral)	Lower extremity lymphedema index (10.13)	6 months
Drobot 2021 [59]	22 (34)	7.3 years	3.1/limb	Circumference reduction (39%)	12 months
Yamamoto 2011 [60] (LE + scrotal edema)	2 (15 & 25)	15 years & 25 years	9 anastomoses (3 sites) & 3 anastomoses (2 sites)	Improved edema	15 months & 53 months

#— “number of”.

## Data Availability

All data presented in this review has been obtained from the references listed below.

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
