# Peer review of "Current Concepts in the Management of Primary Lymphedema"

_medicina, 2023, doi:10.3390/medicina59050894_

Round 1
Reviewer 1 Report
The review focus on the management of primary lymphedema including its treatment methods. The review also includes the diagnosis and variation of primary lymphedema.
The reviewer thinks this is the excellent review about the current management of primary lymphedema. The reviewer thinks the content is rich and concise, there is almost no points for revision.
Author Response
We thank Reviewer #1 for these very kind comments.
Reviewer 2 Report
This review describes the potential mechanisms underlying lymphadema in people of different ages. This brief review describes how genetics and sex of the patient may affect the risk of developing this disease. There are a few sentences that need clarification:
1. Do patients have hyperplastic lymphatics as described above, or hypoplastic as stated here? (pg2)
2. Do patients have hyperplastic lymphatics as described above, or hypoplastic as stated here? (pg 3)
3. Can the authors please provide additional information regarding the role of sex in the development of this disease?
The English was fine. There were areas that needed clarification. Please see the comments
Author Response
We have clarified this section (on page 2) to read: “Unlike primary lymphedema not associated with a vascular malformation in whom lymphatic hypoplasia is most common, lymphatic hyperplasia is the most frequently identified malformation amongst patients with a vascular malformation”
As requested we have further expanded on the role of sex hormones in the development of primary lymphedema in the “incidence and classification” section. A new reference has been added.
Reviewer 3 Report
This is a well-written and comprehensive review which summarizes latest clinic presentation, diagnosis and treatment strategies for primary lymphedema. Because this manuscript is enriched with information, addition of schemas can better guide reader to comprehend all concepts.
Author Response
We thank the reviewer for their kind comments. A flowsheet/schema has been added (Figure 1) that depicts the management strategies for primary lymphedema.
Reviewer 4 Report
Dear Authors
Excellent review on the primary lymphedema management I enjoyed tremendously to read through. But I would like to point out two issues you could accommodate properly to make further improvement of your manuscript on its overall value for the clinicians, as following;
First, you would have been able to provide a bird’s eye view on the ‘primary’ lymphedema as a clinical manifestation of lymphatic malformation (LM) with the vascular malformation point of view, beyond current limited view only as one form of chronic lymphedema. Indeed, the primary lymphedema is more meaningful as one of the vascular malformations as the outcome of defective development along the later stage of lymphangiogenesis, distinctively different from the LM from the earlier stage of lymphangiogenesis, namely ‘lymphangioma’ since both types of the LM infrequently exist together.
Hence, the Introduction part could be further strengthened with proper illustration on this unique position of the primary lymphedema as one of the LMs, which would remain essential part of differential diagnosis. Current contents introduced through ‘Differential Diagnosis’/Introduction can be also incorporated to this aim.
Second, I also have to point out inaccurate(?) interpretation of the recommendation in the consensus document from the International Union of Phlebology 2013 (Ref. 43) you citated as a part of Physiologic Surgery/Microsurgery: “----- surgery performed more than one year following diagnosis increases the risk of failure----” to argue that early surgical intervention may not be necessary, which I wholeheartedly agree with.
But actually, they/UIP Consensus did NOT advocate ‘early’ surgical intervention but advocated ‘no delay’ for the appraisal as the candidates and further surgical intervention if indicated.
Indeed, they recommended “Early-stage lymphedema should be considered as an ideal candidate for reconstructive surgery, since there is a significant risk of from progression of disease to fibrotic changes in the lymphedema patient. There remains significant controversy regarding a suitable waiting period before considering reconstructive surgery. A delay of surgery for more than one year will increase the risk of surgical failure due to chronic lymphatic damage. Therefore, this waiting period should be shortened as much as possible, especially in compliant patients who stand to benefit the most from lymphatic reconstruction.”
They further emphasized with ‘the timing of lymphatic reconstruction remains most crucial and this waiting period should be shortened as much as possible. Thus, indications for lymphatic reconstructive surgery may be summarized as follows: failure to respond to proper therapy at clinical stage I or II; progression of the disease to advanced stages, despite proper treatment; chylous-reflux combined with extremity lymphedema; multiple recurrences of local or systemic infection; poor tolerance- physically, mentally and socioeconomically- of DLT-based conservative treatment’.
Indeed, as you correctly pointed out, unlike secondary lymphedema where patients can be monitored clinically and with angiography to identify disease early in its clinical course, patients with primary lymphedema often are not diagnosed until symptomatic, precluding earlier/pre-clinical intervention.
So, I concur with your claim ‘early surgical intervention may not be necessary’ but agree with the UIP consensus group to advocate ‘shortened waiting period’ before considering reconstructive surgery with ‘no further delay of the appraisal as the surgery candidates and proceed once properly indicated an adjunctive therapy’, because current policy for lymphatic reconstruction that limits the candidates to those who fail DLT-based therapy will result in further damage to the lymphatic system and, subsequently, a higher likelihood of procedure failure. .and recommend though
After all, “reconstructive lymphatic surgery has remained an adjunctive therapy to primary lymphedema. Yet in most cases, when optimally performed, it can result in effective treatment of primary lymphedema.”
All the best,
A Reviewer
Author Response
The aim of our review was moreso to focus on systemic primary lymphedema rather than lymphatic malformations; however, we agree that lymphatic malformations represent a very interesting clinical pathology that is certainly within the spectrum of defective development of the lymphatic system. We have therefore expanded our section on the terminology and relationship between vascular malformation / lymphatic malformations within the section “incidence and classification” as requested.
Further, we have clarified the IUP’s suggestions based on the following quote from their paper:
“There remains significant controversy regarding this waiting period before considering reconstructive surgery. A delay of surgery for more than one year will increase the risk of surgical failure due to chronic lymphatic damage. Therefore, this waiting period should be shortened as much as possible especially in compliant patients who stand to benefit the most from lymphatic reconstruction.”